# Zipformer: A faster and better encoder for automatic speech recognition

**Zengwei Yao, Liyong Guo, Xiaoyu Yang, Wei Kang, Fangjun Kuang,**
**Yifan Yang, Zengrui Jin, Long Lin, Daniel Povey**
Xiaomi Corp., Beijing, China
`dpovey@xiaomi.com`

## Abstract

The Conformer has become the most popular encoder model for automatic speech recognition (ASR). It adds convolution modules to a Transformer to learn both local and global dependencies. In this work we describe a faster, more memory-efficient, and better-performing Transformer, called Zipformer. Modeling changes include: 1) a U-Net-like encoder structure where middle stacks operate at lower frame rates; 2) reorganized block structure with more modules, within which we re-use attention weights for efficiency; 3) a modified form of LayerNorm called BiasNorm allows us to retain some length information; 4) new activation functions SwooshR and SwooshL work better than Swish. We also propose a new optimizer, called ScaledAdam, which scales the update by each tensor's current scale to keep the relative change about the same, and also explictly learns the parameter scale. It achieves faster convergence and better performance than Adam. Extensive experiments on LibriSpeech, Aishell-1, and WenetSpeech datasets demonstrate the effectiveness of our proposed Zipformer over other state-of-the-art ASR models. Our code is publicly available at https://github.com/k2-fsa/icefall.

## 1 Introduction

End-to-end models have achieved remarkable success in automatic speech recognition (ASR). An effective encoder architecture that performs temporal modeling on the speech sequence plays a vital role in end-to-end ASR models. A most prominent example is Conformer (Gulati et al., 2020), which combines the advantages of the convolutional neural network (CNN) models (Zhang et al., 2017; Li et al., 2019; Kriman et al., 2020) and Transformer models (Dong et al., 2018; Karita et al., 2019; Zhang et al., 2020b). By integrating CNN into Transformer (Vaswani et al., 2017), Conformer is able to extract both local and global dependencies on speech sequences, and achieves state-of-the-art performance in ASR.

In this work, we propose a faster, more memory-efficient, and better-performing Transformer as ASR encoder, called *Zipformer*. First, unlike Conformer that operates on the sequence at a constant frame rate, *Zipformer* adopts a U-Net-like (Ronneberger et al., 2015) structure, which consists of multiple stacks downsamping the sequence to various lower frame rates. Second, we re-design the block structure, which is equipped with more modules like two Conformer blocks, and reuses the attention weights for efficiency. We propose *BiasNorm* as a simpler replacement of LayerNorm, which allows for retaining length information in normalization. We also replace Swish with our new activation functions *SwooshR* and *SwooshL* to achieve better results. In addition, we devise a parameter-scale-invariant version of Adam, called *ScaledAdam*, which scales the update by the current parameter scale and also explicitly learns the parameter scale. Compared to Adam, *ScaledAdam* enables faster convergence and better performance.

Extensive experiments are conducted on LibriSpeech, Aishell-1, and WenetSpeech datasets, and results demonstrate the effectiveness of the proposed modeling and optimization-related innovations. *Zipformer* achieves state-of-the-art results on all three datasets. It is worth mentioning that *Zipformer* is the first model ever to achieve results comparable to those reported in the Conformer paper on the LibriSpeech dataset (these results have proved difficult for others to reproduce). In terms of efficiency, *Zipformer* converges faster during training and speeds up the inference by more than

50% compared to previous studies while requiring less GPU memory. We perform detailed ablation studies to investigate the contribution of individual components.

## 2 RELATED WORK

**Model architecture.** Deep convolution architectures have been applied to end-to-end ASR (Zhang et al., 2017; Li et al., 2019). Follow-up works explore improvements by using depthwise separable convolutions (Howard et al., 2017) for efficiency (Kriman et al., 2020), and incorporating squeeze-and-excitation module (Hu et al., 2018) to capture longer context (Han et al., 2020). Inspired by the success of Transformer (Vaswani et al., 2017) in natural language processing (NLP) field, some works adapt Transformer to speech applications (Dong et al., 2018; Karita et al., 2019; Zhang et al., 2020b; Wang et al., 2020; Zhang et al., 2020a). Compared to CNN, the remarkable benefit of Transformer is that it can learn global dependencies based on self-attention, which is essential for speech processing task. By integrating convolution into Transformer, Conformer (Gulati et al., 2020) gains powerful capability of modeling both local and global contextual information, and outperforms all previous ASR models.

Recent works explore architecture changes on Conformer to further reduce the computational cost and improve the recognition performance. Squeezeformer (Kim et al., 2022) adopts a temporal U-Net structure in which the middle modules operate at half frame rates, and also redesigns the block structure to make it similar to the standard Transformer block (Vaswani et al., 2017). Branch-former (Peng et al., 2022) incorporates parallel branches to model various ranged context, in which one branch captures the local context with convolutional gating multi-layer perceptron (MLP), while the other branch learns long-range dependencies with self-attention. E-Branchformer (Kim et al., 2023) further improves Branchformer by enhancing the branch merging mechanism by convolution-based module.

*Zipformer* shares similar ideas about temporal downsampling as the previous work Squeezeformer. However, compared to the fixed downsampling ratio in Squeezeformer, *Zipformer* operates at different downsampling ratios at different encoder stacks and uses much more aggressive downsampling ratios in the middle encoder stacks. In addition to the modeling differences, our work also focuses on optimization-related changes including a new optimizer *ScaledAdam*, which are shown to improve convergence in the experiments.

**End-to-end framework.** Connectionist temporal classification (CTC) (Graves et al., 2006) is one of the earliest frameworks for end-to-end ASR, but its performance is limited by the frame independent assumption. To this end, a hybrid architecture that integrates attention-based encoder-deocder (AED) (Chan et al., 2015) in CTC (Watanabe et al., 2017) (CTC/AED) is proposed to improve the performance. Neural transducer (Graves, 2012), commonly known as RNN-T, addresses the frame independence assumption using a label decoder and a joint network and becomes a popular framework due to its superior performance. Recently, various approaches such as pruning (Kuang et al., 2022; Wang et al., 2023; Mahadeokar et al., 2021) or batch-splitting (Kuchaiev et al., 2019) are proposed to accelerate the training speed and reduce memory usage of neural transducers.

## 3 METHOD

### 3.1 DOWNSAMPLED ENCODER STRUCTURE

Figure 1 presents the overall architecture of the proposed *Zipformer* model. Different from Conformer (Gulati et al., 2020) that processes the sequence at a fixed frame rate of 25Hz, Zipformer uses a U-Net-like structure learning temporal representation at different resolutions in a more efficient way. Specifically, given the acoustic features with frame rate of 100Hz, the convolution-based module called *Conv-Embed* first reduces the length by a factor of 2, resulting in a 50Hz embedding sequence. The obtained sequence is then fed into 6 cascaded stacks to learn temporal representation at frame rates of 50Hz, 25Hz, 12.5Hz, 6.25Hz, 12.5Hz, and 25Hz, respectively. Except for the first stack, the other stacks all adopt the downsampled structures, processing the sequence at lower frame rates. The frame rate between stacks is consistently 50Hz. Different stacks have different embedding dimensions, and the middle stacks have larger dimensions. The output of each stack is truncated or padded with zeros to match the dimension of the next stack. The final encoder output dimension is set to the maximum of all stacks' dimensions. Specifically, if the last stack output has the largest dimension, it is taken as the encoder output; otherwise, it is concatenated from differ-

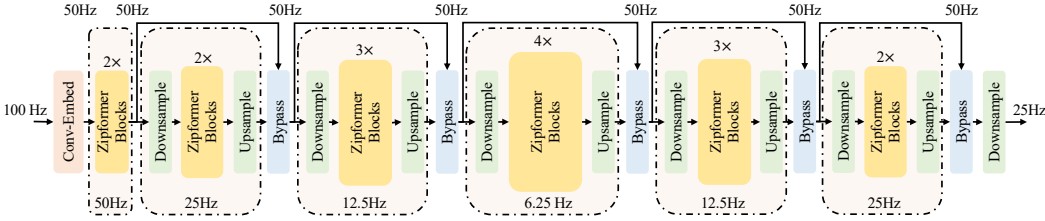

Figure 1: Overall architecture of Zipformer.

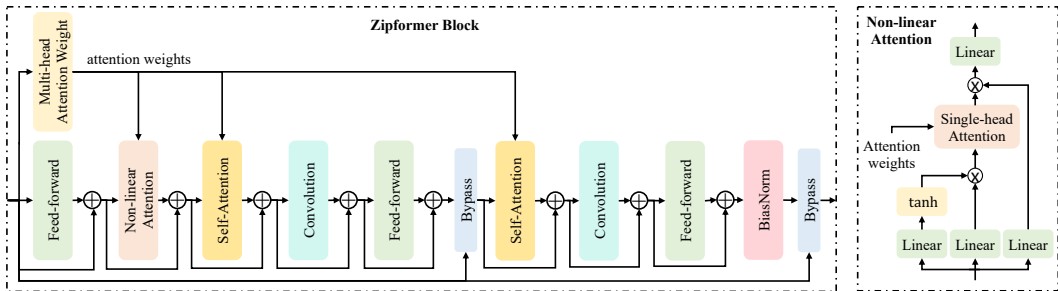

Figure 2: (Left): Zipformer block structure. (Right): Non-Linear Attention module structure.

ent pieces of stack outputs, taking each dimension from the most recent output that has it present. Finally, a *Downsample* module converts the sequence to 25Hz, resulting in the encoder output.

**Conv-Embed.** In *Conv-Embed* we use three 2-D convolutional layers with time × frequency strides of $1 \times 2$, $2 \times 2$, and $1 \times 2$, and output channels of 8, 32, and 128, respectively. Subsequently, we utilize one ConvNeXt layer (Liu et al., 2022) similar to Nextformer (Jiang et al., 2022), which is composed of a depth-wise convolution with kernel size of $7 \times 7$, a point-wise convolution with 384 output channels, a *SwooshL* activation function (described in Section 3.4), and a point-wise convolution with 128 output channels. Residual connection is applied on the ConvNeXt module. Finally, a linear layer followed by a *BiasNorm* (described in Section 3.3) is used to adjust the feature dimension to match the first stack.

**Downsampled stacks.** In the downsampled stacks, the pairwise *Downsample* and *Upsample* modules perform symmetric scaling down and scaling up in sequence length, respectively, using almost the simplest methods. For example, with a factor of 2, the *Downsample* module averages every 2 frames with 2 learnable scalar weights (after softmax normalization), and the *Upsample* module just repeats each frame twice. After downsampling, it employs the stacking *Zipformer* blocks (described in Section 3.2) for temporal modeling at lower frame rates. Finally, it utilizes the *Bypass* module (described in Section 3.2) to combine the stack input and stack output in a learnable way.

## 3.2 ZIPFORMER BLOCK

Conformer block consists of four modules: feed-forward, Multi-Head Self-Attention (MHSA), convolution, and feed-forward. MHSA learns global context by two steps: computing attention weights using the dot-product operation and aggregating different frames with these attention weights. However, MHSA typically accounts for a large computational cost, since above two steps both require quadratic complexity with respect to the sequence length. Hence, we decompose MHSA into two individual modules according to above two steps: Multi-Head Attention Weight (*MHAW*) and Self-Attention (*SA*). This change allows to perform the attention computation twice more efficiently in each block by using one *MHAW* module and two *SA* modules. In addition, we propose a new module called Non-Linear Attention (*NLA*) to make full use of the computed attention weights to capture the global information.

As illustrated in Figure 2 (Left), *Zipformer* block is equipped with about twice the depth of the Conformer block (Gulati et al., 2020). The main motivation is to allow the re-use of the attention weights to save time and memory. Specifically, the block input is first fed into an *MHAW* module, which calculates the attention weights and shares them with an *NLA* module and two *SA* modules. Meanwhile, the block input is also fed into a feed-forward module followed by the *NLA* module.

Then it applies two module groups, each consisting of *SA*, convolution, and feed-forward. Finally, a *BiasNorm* (described in Section 3.3) is used to normalize the block output. In addition to the regular residual connections using adding operation, each block utilizes two *Bypass* modules to combine the block input and the module outputs, placed in the middle and end of the block. Note that different from regular Transformer models (Vaswani et al., 2017), we don't use normalization layer such as LayerNorm (Ba et al., 2016) for each module to periodically prevent activations from becoming either too large or too small, since our proposed *ScaledAdam* optimizer is able to learn the parameter scales (described in Section 3.5).

**Non-Linear Attention.** Figure 2 (Right) presents the *NLA* structure. It also leverages the pre-computed attention weights from *MHAW* to aggregate the embedding vectors over the time axis, which is similar to *SA*. Specifically, it first projects the input with 3 linear layers to $A$, $B$, and $C$, each being of 3/4 input dimension. The module output is $\text{linear}(A \odot \text{attention}(\tanh(B) \odot C))$, where $\odot$ denotes the element-wise multiplication, $\text{attention}$ represents matrix-multiplying on the time axis by a single head of previously computed attention weights, and the linear layer recovers the dimension to the same as the input.

**Bypass.** The *Bypass* module learns channel-wise scalar weights $\mathbf{c}$ to combine the module input $\mathbf{x}$ and module output $\mathbf{y}$: $(1 - \mathbf{c}) \odot \mathbf{x} + \mathbf{c} \odot \mathbf{y}$. In training, we initially limit the values of $\mathbf{c}$ in range of $[0.9, 1.0]$ and then change the minimum to 0.2 after 20000 steps. We found that making modules "straight-through" at the beginning (i.e. allowing very little bypass) helps model convergence..

## 3.3 BIASNORM

Conformer (Gulati et al., 2020) utilizes LayerNorm (Ba et al., 2016) to normalize the module activations. Given $\mathbf{x}$ with $D$ channels, LayerNorm is formulated as:

$$\text{LayerNorm}(\mathbf{x}) = \frac{\mathbf{x} - \text{E}[\mathbf{x}]}{\sqrt{\text{Var}[\mathbf{x}] + \epsilon}} \odot \boldsymbol{\gamma} + \boldsymbol{\beta}. \tag{1}$$

Specifically, it first computes the mean $\text{E}[\mathbf{x}]$ and the standard-deviation $\sqrt{\text{Var}[\mathbf{x}]}$ for normalizing, scaling the vector length to $\sqrt{D}$. Then it uses the learnable channel-wise scale $\boldsymbol{\gamma}$ and bias $\boldsymbol{\beta}$ for transformation, which helps to adjust the size of activations and balance the relative contributions of specific modules. However, we observe that the trained Conformer using LayerNorm suffers from two problems: 1) It sometimes sets one channel to a large constant value, e.g. 50. We argue that it aims to "defeat" the LayerNorm which fully removes the vector length, functioning as a very large value so that length information could be retained after normalization. 2) Some modules (typically feed-forward or convolution) are "dead" as they have extremely small output values, e.g., $10^{-6}$. We argue that early in training, the un-trained modules are not useful so they are "turned off" by the LayerNorm scale $\boldsymbol{\gamma}$ approaching zero. If the scale $\boldsymbol{\gamma}$ oscillates around zero, the inconsistent sign constantly reverses the gradient directions back-propagating to the modules. Because of the inconsistent gradient sign, the modules never learn anything useful, since this is a bad local optimum which is hard to escape because of the dynamics of stochastic gradient descent-like updates.

To address above problems, we propose the *BiasNorm* which is intended to be a simpler replacement of LayerNorm. Specifically, *BiasNorm* is formulated as:

$$BiasNorm(\mathbf{x}) = \frac{\mathbf{x}}{\text{RMS}[\mathbf{x} - \mathbf{b}]} \cdot \exp(\gamma), \tag{2}$$

where $\mathbf{b}$ is the learnable channel-wise bias, $\text{RMS}[\mathbf{x} - \mathbf{b}]$ is the root-mean-square value taken over channels, and $\gamma$ is a scalar. We first remove the operation of mean subtraction since it is a waste of time unless it follows a non-linearity. The bias $\mathbf{b}$ serves as the large constant value which allows to retain the vector length information after normalization. Since the scale $\exp(\gamma)$ is always positive, it avoids the gradient oscillation problem.

## 3.4 SWOOSHR AND SWOOSHL ACTIVATION FUNCTIONS

Conformer (Gulati et al., 2020) adopts Swish (Ramachandran et al., 2017) activation function with the following formula:

$$\text{Swish}(x) = x \cdot (1 + \exp(-x))^{-1}. \tag{3}$$

In this work, we propose two new activation functions respectively called *SwooshR* and *SwooshL* as replacements of Swish:

$$SwooshR(x) = \log(1 + \exp(x-1)) - 0.08x - 0.313261687,$$
$$SwooshL(x) = \log(1 + \exp(x-4)) - 0.08x - 0.035. \tag{4}$$

In *SwooshR*, the offset 0.313261687 is to make it pass through the origin; in *SwooshL*, the offset 0.035 was tuned, which slightly outperformed the value exactly making the curve pass through the origin. We present the curves of Swish, *SwooshR*, and *SwooshL* in Appendix Section A.2. *SwooshL* is roughly a right shifted version of *SwooshR*. Note that the suffix "L" or "R" represents whether the left or right zero-crossing is at or around $x = 0$. Similar to Swish, *SwooshR* and *SwooshL* have lower bounds and are non-monotonic. Compared to Swish, the most striking difference is that *SwooshR* and *SwooshL* have non-vanishing slopes for negative inputs, which helps to escape from situations where the input is always negative and prevents the denominator term in Adam-type updates from getting dangerously small. When replacing Swish with *SwooshR*, we observe that the modules with bypass connections, such as feed-forward and ConvNeXt, tend to learn a large negative bias in the preceding linear layer to learn "normally-off" behavior. Therefore, we use *SwooshL* for these "normally-off" modules and use *SwooshR* for convolution modules and the rest of *Conv-Embed*.

### 3.5 SCALEDADAM OPTIMIZER

We propose a parameter-scale-invariant version of Adam (Kingma & Ba, 2014) called *ScaledAdam*, which enables faster convergence and better performance. *ScaledAdam* scales each parameter's update proportional to the scale of that parameter, and also explicitly learns the parameter scale. Algorithm 1 in Appendix Section A.1.1 presents the pseudo-code of the *ScaledAdam*.

Let $f(\boldsymbol{\theta})$ be the loss function that we aim to minimize, which is differentiable w.r.t. the learnable parameter $\boldsymbol{\theta}$. At each step $t$, Adam computes the parameter gradient $\mathbf{g}_t = \nabla_{\boldsymbol{\theta}} f(\boldsymbol{\theta}_{t-1})$, and updates the first moment $\mathbf{m}_t = \beta_1 \cdot \mathbf{m}_{t-1} + (1-\beta_1) \cdot \mathbf{g}_t$ and the second moment $\mathbf{v}_t = \beta_2 \cdot \mathbf{v}_{t-1} + (1-\beta_2) \cdot \mathbf{g}_t^2$ of gradients, where $\beta_1, \beta_2 \in [0, 1)$ are coefficients used to compute the moving averages. The parameter update $\boldsymbol{\Delta}_t$ at step $t$ is formulated as:

$$\boldsymbol{\Delta}_t = -\alpha_t \cdot \frac{\sqrt{1-\beta_2^t}}{1-\beta_1^t} \cdot \frac{\mathbf{m}_t}{\sqrt{\mathbf{v}_t} + \epsilon}, \tag{5}$$

where $\alpha_t$ is the learning rate typically specified by an external schedule, $\frac{\sqrt{1-\beta_2^t}}{1-\beta_1^t}$ is the bias-correction term, and $\epsilon = 10^{-8}$. Whilst Adam is invariant to gradient scale of each parameter, we argue that it still suffers from two limitations: 1) The update $\boldsymbol{\Delta}_t$ in Equation 5 does not take into account the parameter scale (denoted as $r_{t-1}$). Considering the relative parameter change $\boldsymbol{\Delta}_t/r_{t-1}$, Adam might cause learning in relative terms too slowly for parameters with large scales, or too fast for parameters with small scales. 2) It is difficult to learn the parameter scale directly, as the direction of growing or shrinking the parameter tensor is a very specific direction in a large-dimensional space. It's particularly difficult to shrink a parameter, since each gradient step $\mathbf{g}_t$ adds noise which tends to grow the parameter norm.

**Scaling update.** To keep the relative change $\boldsymbol{\Delta}_t/r_{t-1}$ over parameters of varying scales about the same, we scale the update $\boldsymbol{\Delta}_t$ in Equation 5 by the parameter scale $r_{t-1}$:

$$\boldsymbol{\Delta}_t' = -\alpha_t \cdot r_{t-1} \cdot \frac{\sqrt{1-\beta_2^t}}{1-\beta_1^t} \cdot \frac{\mathbf{m}_t}{\sqrt{\mathbf{v}_t} + \epsilon}. \tag{6}$$

We compute the parameter scale $r_{t-1}$ as the root-mean-square value $\text{RMS}[\boldsymbol{\theta}_{t-1}]$. Because the ScaledAdam update is less prone to divergence than Adam, we use a learning rate schedule called *Eden* that does not have a long warm-up period; we also use absolutely larger learning rate values because the parameter RMS value is normally much less than one.

**Learning parameter scale.** To explicitly learn the parameter scale, we treat it as a regular parameter to be learned, as if we have factored each parameter as $\boldsymbol{\theta} = r \cdot \boldsymbol{\theta}'$, and we are doing gradient descent on the parameter scale $r$ and the underlying parameter $\boldsymbol{\theta}'$. Let $h$ be the gradient of the parameter scale $r$, at step $t$ we get $h_t = \nabla_r f(\boldsymbol{\theta}_{t-1}) = \mathbf{g}_t \cdot \boldsymbol{\theta}'_{t-1}$. Since Adam is nearly invariant to changes in the gradient scale, for simplicity we replace this with $h_t = \mathbf{g}_t \cdot (r_{t-1} \odot \boldsymbol{\theta}'_{t-1}) = \mathbf{g}_t \cdot \boldsymbol{\theta}_{t-1}$. Following the Adam algorithm, we maintain the first moment $n_t = \beta_1 \cdot n_{t-1} + (1-\beta_1) \cdot h_t$ and the second moment $w_t = \beta_2 \cdot w_{t-1} + (1-\beta_2) \cdot h_t^2$ of the scale gradients $h_t$. The parameter change on $\boldsymbol{\theta}$ caused

by updating parameter scale from $r_{t-1}$ to $r_t$ is $\boldsymbol{\Delta}'_{t,r} = (r_t - r_{t-1}) \odot \boldsymbol{\theta}'_{t-1}$. Similar to Equation 6, we also integrate the parameter scale $r_{t-1}$ into the update $\boldsymbol{\Delta}'_{t,r}$:

$$
\begin{aligned}
\boldsymbol{\Delta}'_{t,r} &= -\eta \cdot \alpha_t \cdot r_{t-1} \cdot \frac{\sqrt{1-\beta_2^t}}{1-\beta_1^t} \cdot \frac{n_t}{\sqrt{w_t}+\epsilon} \odot \boldsymbol{\theta}'_{t-1} \\
&= -\eta \cdot \alpha_t \cdot \frac{\sqrt{1-\beta_2^t}}{1-\beta_1^t} \cdot \frac{n_t}{\sqrt{w_t}+\epsilon} \odot \boldsymbol{\theta}_{t-1}.
\end{aligned}
\tag{7}
$$

where $\eta$ is a scaling factor on learning rate $\alpha_t$, and we found that setting $\eta = 0.1$ helps to stabilize the training. Now the update $\boldsymbol{\Delta}'_t$ is replaced with $\boldsymbol{\Delta}'_{t,r} + \boldsymbol{\Delta}'_t$, which amounts to adding an extra gradient term in the direction of growing or shrinking each parameter. This also allows to simplify the network structure by removing most of normalization layers in our *Zipformer* Block (described in Section 3.2), since the modules now can easily learn to scale the activations in a suitable range. One similar method called weight normalization (Salimans & Kingma, 2016) decouples the parameter norm from its direction to speed up the convergence. It replaces each parameter with two parameters, respectively specifying the direction and the magnitude. However, ScaledAdam learns the parameter scales by adding an extra update term $\boldsymbol{\Delta}'_{t,r}$, which makes writing the modeling code simpler.

**Eden schedule.** The proposed *Eden* learning rate schedule is formulated as:

$$
\alpha_t = \alpha_{\text{base}} \cdot \left(\frac{t^2 + \alpha_{\text{step}}^2}{\alpha_{\text{step}}^2}\right)^{-0.25} \cdot \left(\frac{e^2 + \alpha_{\text{epoch}}^2}{\alpha_{\text{epoch}}^2}\right)^{-0.25} \cdot \text{linear}(\alpha_{\text{start}}, t_{\text{warmup}}, t).
\tag{8}
$$

Herein, $t$ is the step index, $e$ is the epoch index, $\alpha_{\text{step}}$ and $\alpha_{\text{epoch}}$ respectively control the number of steps and number of epochs after which we start significantly decreasing the learning rate, $\text{linear}(\alpha_{\text{start}}, t_{\text{warmup}}, t)$ is a warmup scale increasing linearly from $\alpha_{\text{start}}$ to 1 over $t_{\text{warmup}}$ steps and then staying constant at 1, $\alpha_{\text{base}}$ is the maximum value when setting $\alpha_{\text{start}} = 1, \alpha_{\text{warmup}} = 0$. The reason for making *Eden* dependent on both the step index $t$ and the epoch index $e$ is to keep the amount of parameter change after certain amount of training data (e.g., one hour) approximately constant when we change the batch size, so the schedule parameters should not have to be re-tuned if we change the batch size. Other versions of *Eden* replace the "epoch" parts of the formula with some suitable measure of the amount of data seen. In this work, we use $\alpha_{\text{base}} = 0.045, \alpha_{\text{start}} = 0.5$, and $t_{\text{warmup}} = 500$.

**Efficient implementation.** To speedup the optimization in *ScaledAdam*, we group the parameters into batches according to their shape and perform the computation batch by batch. Note that this doesn't affect the outcome. *ScaledAdam* just requires a little more memory than Adam to cache the gradient moments $n_t$ and $w_t$ (in Equation 7) for the parameter scales.

## 4 EXPERIMENTS

### 4.0.1 EXPERIMENTAL SETUP

**Architecture variants.** We build our *Zipformer* variants with three model scales: small (*Zipformer*-S), medium (*Zipformer*-M), and large (*Zipformer*-L). For the 6 encoder stacks, the numbers of attention heads are set to {4,4,4,8,4,4}, the convolution kernel sizes are set to {31,31,15,15,15,31}. In each attention head, the query dimension and value dimension are set to 32 and 12, respectively. For the three feed-forward modules in each *Zipformer* block, the hidden dimensions in the first one and the last one are 3/4 and 5/4 of that in the middle one. We adjust the layers numbers, the embedding dimensions, and the hidden dimensions of the middle feed-forward in each stack to obtain different model scales:

Table 1: Configuration of *Zipformer* at three different scales.

| Scale | layer-numbers | embedding-dimensions | feed-forward-dimensions |
|-------|---------------|----------------------|-------------------------|
| S | {2,2,2,2,2,2} | {192,256,256,256,256,256} | {512,768,768,768,768,768} |
| M | {2,2,3,4,3,2} | {192,256,384,512,384,256} | {512,768,1024,1536,1024,768} |
| L | {2,2,4,5,4,2} | {192,256,512,768,512,256} | {512,768,1536,2048,1536,768} |

**Datasets.** We perform experiments to compare our *Zipformer* with state-of-the-other models on three open-source datasets: 1) LibriSpeech (Panayotov et al., 2015) which consists of about 1000

hours of English audiobook reading; 2) Aishell-1 (Bu et al., 2017) which contains 170 hours of Mandarin speech; 3) WenetSpeech (Zhang et al., 2022a) which consists of 10000+ hours of multi-domain Mandarin speech.

**Implementation details.** We use Lhotse (Żelasko et al., 2021) toolkit for speech data preparation. The model inputs are 80-dimension Mel filter-bank features extracted on 25ms frames with frame shift of 10ms. Speed perturbation (Ko et al., 2015) with factors of 0.9, 1.0, and 1.1 is used to augment the training data. SpecAugment (Park et al., 2019) is also applied during training. We use mixed precision training for our *Zipformer* models. We also employ the activation constraints including *Balancer* and *Whitener* to ensure training consistency and stability. The details of *Balancer* and *Whitener* are presented in Appendix Section A.3. Pruned transducer (Kuang et al., 2022), a memory-efficient version of transducer loss that prunes path with minor posterior is used as the training objective. During decoding, beam search of size 4 with the constraint of emitting at most one symbol per frame is employed (Kang et al., 2023). We don't use external language models for rescoring, since in this work we focus on improving the encoder model. We employ word-error-rate (WER) and character error rate (CER) as evaluation metric for English and Mandarin datasets, respectively. By default, all of our models are trained on 32GB NVIDIA Tesla V100 GPUs. For Librispeech dataset, *Zipformer*-M and *Zipformer*-L are trained for 50 epochs on 4 GPUs, and *Zipformer*-S is trained for 50 epochs on 2 GPUs. For Aishell-1 dataset, our models are trained for 56 epochs on 2 GPUs. For WenetSpeech dataset, our models are trained for 14 epochs on 4 GPUs.

### 4.0.2 COMPARISON WITH STATE-OF-THE-ART MODELS

In this section, we compare the proposed Zipformer with other state-of-the-art models.

**LibriSpeech dataset.** Table 2 shows the results on LibriSpeech test datasets for *Zipformer* and other state-of-the-art models. For Conformer, we also list the WERs reproduced by us and other open-source frameworks. Note that there is a performance gap between the open-source reproduced Conformer and the original Conformer. Our *Zipformer*-S model achieves lower WERs than all variants of Squeezeformer while having much fewer parameters and floating point operations (FLOPs). Our *Zipformer*-L outperforms Squeezeformer-L, Branchformer and our reproduced Conformer-L by a large margin while saving over 50% FLOPs. Noticeably, when trained on 8 80G NVIDIA Tesla A100 GPUs for 170 epochs, *Zipformer*-L achieves WERs of 2.00%/4.38% with sufficient computing resources (last row), which is the first model to approach Conformer-L to the best of our knowledge.

We also compare the speed and memory usage between the proposed *Zipfomer* and other state-of-the-art models. Figure 3 presents the comparison results in terms of averaged inference time and peak memory usage in inference mode for batches of 30-second audios on an NVIDIA Tesla V100 GPU. The batch size is set to 30 to ensure all models do not have out of memory problems during inference. In overall, *Zipformer* models achieves better trade-off between performance and efficiency than other models. Especially for the large scale, *Zipformer*-L requires much less computation time and memory than other counterparts.

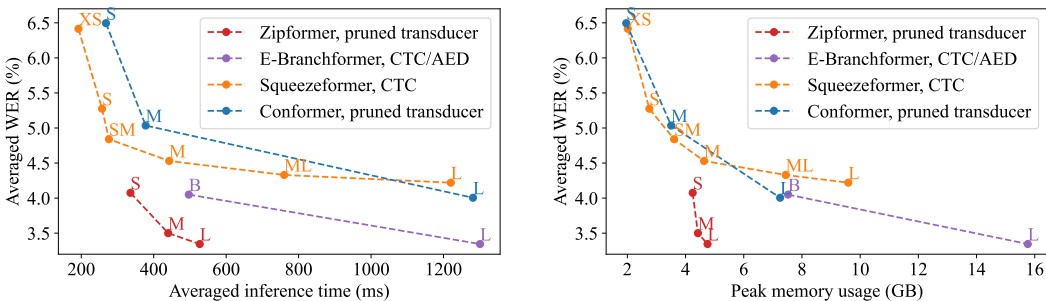

Figure 3: (Left) Averaged inference time and (Right) peak memory usage vs. WER comparison for different models. The WER is averaged on LibriSpeech *test-clean* and *test-other*. Averaged inference time and peak memory usage are reported for the **encoders** in inference mode for batches of 30-second audios with batch size of 30 on a single NVIDIA Tesla V100 GPU.

**Aishell-1 dataset.** Table 3 shows the CERs on Aishell-1 dataset. Compared to the Conformer model implemented in ESPnet toolkit, our *Zipformer*-S achieves better performance with fewer parameters. Scaling up the model leads to lower WERs, and *Zipformer*-M/L outperform all other models.

Table 2: WER(%) comparison between different models on LibriSpeech dataset. We also include the number of parameters and FLOPs of encoder for a 30s input audio measured with DeepSpeed (Rasley et al., 2020). *Trained with 8 80G NVIDIA Tesla A100 GPUs for 170 epochs.

| Model | Type | Params (M) | GFLOPs | *test-clean* (%) | *test-other* (%) |
|---|---|---|---|---|---|
| Squeezeformer-XS (Kim et al., 2022) | CTC | 9.0 | 18.2 | 3.74 | 9.09 |
| Squeezeformer-S (Kim et al., 2022) | CTC | 18.6 | 33.7 | 3.08 | 7.47 |
| Squeezeformer-SM (Kim et al., 2022) | CTC | 28.2 | 47.6 | 2.79 | 6.89 |
| Squeezeformer-M (Kim et al., 2022) | CTC | 55.6 | 88.4 | 2.56 | 6.50 |
| Squeezeformer-ML (Kim et al., 2022) | CTC | 125.1 | 183.3 | 2.61 | 6.05 |
| Squeezeformer-L (Kim et al., 2022) | CTC | 236.3 | 333.7 | 2.47 | 5.97 |
| E-Branchformer-B (Kim et al., 2023) | CTC/AED | 41.1 | 78.1 | 2.49 | 5.61 |
| Branchformer (Peng et al., 2022) | CTC/AED | 116.2 | 238.3 | 2.4 | 5.5 |
| E-Branchformer-L (Kim et al., 2023) | CTC/AED | 148.9 | 284.4 | 2.14 | 4.55 |
| Conformer-S (Gulati et al., 2020) | transducer | 10.3 | – | 2.7 | 6.3 |
| Conformer-M (Gulati et al., 2020) | transducer | 30.7 | – | 2.3 | 5.0 |
| Conformer-L (Gulati et al., 2020) | transducer | 118.8 | – | 2.1 | **4.3** |
| Conformer in WeNet (Zhang et al., 2022b) | CTC/AED | 121.3 | – | 2.66 | 6.53 |
| Conformer in ESPnet (Miyazaki et al., 2023) | CTC/AED | 113.2 | – | 2.29 | 5.13 |
| Conformer-S | pruned transducer | 9.8 | 29.1 | 3.75 | 9.24 |
| Conformer-M | pruned transducer | 28.4 | 77.0 | 2.96 | 7.11 |
| Conformer-L | pruned transducer | 122.5 | 294.2 | 2.46 | 5.55 |
| *Zipformer*-S | pruned transducer | 23.3 | 40.8 | 2.42 | 5.73 |
| *Zipformer*-M | pruned transducer | 65.6 | 62.9 | 2.21 | 4.79 |
| *Zipformer*-L | pruned transducer | 148.4 | 107.7 | 2.06 | 4.63 |
| *Zipformer*-L* | pruned transducer | 148.4 | 107.7 | **2.00** | 4.38 |

**WenetSpeech.** Table 4 presents the experimental results on WenetSpeech dataset. Again, our *Zipformer*-M and *Zipformer*-L outperform all other models on *Test_Net* and *Test_Meeting* test sets. With only one third of the parameters, our *Zipformer*-S yields lower WERs than Conformer models.

Table 3: CER(%) comparison between different models on Aishell-1 dataset.

| Model | Params (M) | Type | Dev | Test |
|---|---|---|---|---|
| Conformer in ESPnet (Watanabe et al., 2018) | 46.2 | CTC/AED | 4.5 | 4.9 |
| Conformer in WeNet (Yao et al., 2021) | 46.3 | CTC/AED | – | 4.61 |
| E-Branchformer in ESPnet (Watanabe et al., 2018) | 37.9 | CTC/AED | 4.2 | 4.5 |
| Branchformer (Peng et al., 2022) | 45.4 | CTC/AED | 4.19 | 4.43 |
| *Zipformer*-S | 30.2 | pruned transducer | 4.4 | 4.67 |
| *Zipformer*-M | 73.4 | pruned transducer | 4.13 | 4.4 |
| *Zipformer*-L | 157.3 | pruned transducer | **4.03** | **4.28** |

Table 4: CER(%) comparison between different models on WenetSpeech dataset.

| Model | Params (M) | Type | *Dev* | *Test_Net* | *Test_Meeting* |
|---|---|---|---|---|---|
| Conformer in ESPnet (Watanabe et al., 2018) | 116.9 | CTC/AED | 9.70 | 8.90 | 15.90 |
| Conformer in WeNet (Yao et al., 2021) | 116.9 | CTC/AED | 8.88 | 9.70 | 15.59 |
| Conformer-MoE(16e) (You et al., 2022) | 425 | CTC/AED, MoE | 7.67 | 8.28 | 13.96 |
| Conformer-MoE(32e) (You et al., 2022) | – | CTC/AED, MoE | 7.49 | 7.99 | 13.69 |
| Conformer-MoE(64e) (You et al., 2022) | – | CTC/AED, MoE | **7.19** | 8.36 | 13.72 |
| *Zipformer*-S | 32.3 | pruned transducer | 7.96 | 8.6 | 13.97 |
| *Zipformer*-M | 75.9 | pruned transducer | 7.32 | 7.61 | 12.35 |
| *Zipformer*-L | 160.9 | pruned transducer | 7.29 | **7.24** | **12.06** |

### 4.0.3 ABLATION STUDIES

We perform ablation experiments on LibriSpeech dataset to investigate the effect of each proposed functional technique. With *Zipformer*-M as the base model, we make one change each time while keeping the others untouched. Table 5 presents the experimental results.

**Encoder structure.** We remove the temporal downsampling structure from *Zipformer* and use *Conv-Embed* with downsampling rate of 4 like Conformer. The resulting model has 12 *Zipformer* blocks with a constant embedding dimension of 512 and has more parameters than the base model.

Table 5: Ablation studies for *Zipformer*-M, including encoder structure, block structure, normalization layer, activation function and optimizer.

| Ablation | Params (M) | test-clean (%) | test-other (%) |
|---|---|---|---|
| *Zipformer*-M | 65.6 | 2.21 | 4.79 |
| **Encoder structure** | | | |
| No temporal downsampling | 94.2 | 2.23 | 5.09 |
| **Block structure** | | | |
| Double Conformer-style blocks | 73.9 | 2.18 | 4.95 |
| No *NLA* | 58.7 | 2.16 | 4.97 |
| No *NLA*, no attention weights sharing | 60.9 | 2.20 | 5.10 |
| No *Bypass* | 65.5 | 2.25 | 4.86 |
| **Normalization layer** | | | |
| LayerNorm | 65.6 | 2.29 | 4.97 |
| **Activation function** | | | |
| Only *SwooshR* | 65.5 | 2.32 | 5.21 |
| Swish | 65.5 | 2.27 | 5.37 |
| **Optimizer** | | | |
| Adam | 65.6 | 2.38 | 5.51 |

Experimental results in Table 5 show that the resulting model without the downsampled structure yields higher WERs on both test set. It indicates that the temporal downsampling structure for efficiency does not cause information loss, but facilitates the modeling capacity with less parameters.

**Block structure.** As each *Zipformer* block has roughly twice modules as a Conformer block, we replace each *Zipformer* block in the base model with two Conformer blocks stacked together. This leads to 0.16% absolute WER reduction on *test-other* even with a larger model size, suggesting the benefits of *Zipformer* block structure. Removing either *NLA* or *Bypass* leads to performance degradation. If we further remove the attention weights sharing mechanism after removing *NLA*, the model has slightly more parameters and slower inference speed, but the WERs are not improved. We hypothesize that the two attention weights inside one *Zipformer* block are quite consistent and sharing them does not harm the model.

**Normalization layer.** Replacing *BiasNorm* with LayerNorm in *Zipformer* leads to WER drops of 0.08% and 0.18% on *test-clean* and *test-other*, respectively. It indicates the advantage of the proposed *BiasNorm* which allows to retain some length information in normalization.

**Activation function.** When using only *SwooshR* for all modules in *Zipformer*, the WER drops by 0.11% and 0.42% on *test-clean* and *test-other*, respectively, which validates the effectiveness of particularly using *SwooshL* for the "normally-off" modules. Employing Swish leads to more performance degradation, which indicates the advantage of *SwooshR* over Swish.

**Optimizer.** When using Adam to train *Zipformer*, we have to apply *BiasNorm* for each module in *Zipformer* block to avoid model divergence, since Adam cannot learn the scale of each parameter to adjust the module activations like *ScaledAdam*. We try different learning rate factors (denoted as $\alpha_{\text{base}}$) for *ScaledAdam* (0.025, 0.035, 0.045, 0.055) and Adam (2.5, 5.0, 7.5, 10.0) separately. Following (Gulati et al., 2020), the learning rate schedule for Adam is $\alpha_t = \alpha_{\text{base}} \cdot 512^{-0.5} \cdot \min(t^{-0.5}, t \cdot 10000^{-1.5})$. Figure A.2 in Appendix Section A.1.2 presents the averaged WERs on *test-clean* and *test-other* at different epochs as well as the learning rates at different steps. We show the best results of *ScaledAdam* with $\alpha_{\text{base}} = 0.045$ and Adam with $\alpha_{\text{base}} = 7.5$ in Table 5. ScaledAdam outperforms Adam by 0.17% and 0.72% on *test-clean* and *test-other*, respectively. The results indicate that ScaledAdam enables faster convergence and better performance than Adam.

## 5 CONCLUSION

In this work, we present the *Zipformer*, which serves as an efficient ASR encoder. It has an U-Net-like encoder structure, which downsamples the sequence to various lower frame rates. The re-designed block structure equipped with more modules reuses the computed attention weights for efficiency. It also employs the new normalization method *BiasNorm*, as well as the new activation functions *SwooshR* and *SwooshL*. Meanwhile, the proposed optimizer *ScaledAdam* enables faster convergence and better performance. Extensive experiments on LibriSpeech, Aishell-1 and Wenet-Speech datasets have demonstrated the effectiveness of the proposed *Zipformer*.

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

# A APPENDIX

## A.1 SCALEDADAM OPTIMIZER

### A.1.1 SCALEDADAM ALGORITHM.

---

**Algorithm 1** *ScaledAdam* Algorithm. RMS refers to root-mean-square function. $g_t^2$ refers to $g_t \odot g_t$. $\alpha_t$ is controlled by Eden learning rate schedule. Good default settings are $\beta_1 = 0.9, \beta_2 = 0.98, \eta = 0.1$, and $\epsilon = 10^{-8}$.

---

**Require:** learning rate $\alpha_t$; exponential decay rates for the moment estimates $\beta_1, \beta_2 \in [0, 1)$; scaling factor on the learning rate for parameter scale $\eta$; objective function $f(\boldsymbol{\theta})$ with parameters $\boldsymbol{\theta}$; initial parameter $\boldsymbol{\theta}_0$.

$\quad t \leftarrow 0$ ▷ Initialize step.

$\quad \mathbf{m}_0 \leftarrow 0, \mathbf{v}_0 \leftarrow 0$ ▷ Initialize first and second moment of parameter gradient.

$\quad n_0 \leftarrow 0, w_0 \leftarrow 0$ ▷ Initialize first and second moments of parameter scale gradient.

$\quad r_0 \leftarrow \mathrm{RMS}(\boldsymbol{\theta}_0)$ ▷ Initialize parameter scale.

$\quad$ **while** $\boldsymbol{\theta}_t$ not converged **do**

$\quad\quad t \leftarrow t + 1$

$\quad\quad \mathbf{g}_t \leftarrow \nabla_{\boldsymbol{\theta}} f_t(\boldsymbol{\theta}_{t-1})$ ▷ Get parameter gradient.

$\quad\quad h_t \leftarrow \mathbf{g}_t \cdot \boldsymbol{\theta}_{t-1}$ ▷ Get parameter scale gradient.

$\quad\quad r_{t-1} \leftarrow \mathrm{RMS}(\boldsymbol{\theta}_{t-1})$ ▷ Update the parameter scale.

$\quad\quad \mathbf{m}_t = \beta_1 \cdot \mathbf{m}_{t-1} + (1 - \beta_1) \cdot \mathbf{g}_t$ ▷ Update first moment of parameter gradient.

$\quad\quad \mathbf{v}_t = \beta_2 \cdot \mathbf{v}_{t-1} + (1 - \beta_2) \cdot \mathbf{g}_t^2$ ▷ Update second moment of parameter gradient.

$\quad\quad \boldsymbol{\Delta}'_t = -\alpha_t \cdot r_{t-1} \cdot \frac{\sqrt{1-\beta_2^t}}{1-\beta_1^t} \cdot \frac{\mathbf{m}_t}{\sqrt{\mathbf{v}_t}+\epsilon}$ ▷ Compute parameter change.

$\quad\quad n_t \leftarrow \beta_1 \cdot n_{t-1} + (1 - \beta_1) \cdot h_t$ ▷ Update first moment of parameter scale gradient.

$\quad\quad w_t \leftarrow \beta_2 \cdot w_{t-1} + (1 - \beta_2) \cdot h_t^2$ ▷ Update second moment of parameter scale gradient.

$\quad\quad \boldsymbol{\Delta}'_{t,r} \leftarrow -\eta \cdot \alpha_t \cdot \frac{\sqrt{1-\beta_2^t}}{1-\beta_1^t} \cdot \frac{n_t}{\sqrt{w_t}+\epsilon} \odot \boldsymbol{\theta}_{t-1}$ ▷ Compute parameter change by updating parameter scale.

$\quad\quad \boldsymbol{\theta}_t \leftarrow \boldsymbol{\theta}_{t-1} + \boldsymbol{\Delta}'_t + \boldsymbol{\Delta}'_{t,r}$ ▷ Update parameter.

$\quad$ **end while**

---

### A.1.2 COMPARISON BETWEEN SCALEDADAM AND ADAM.

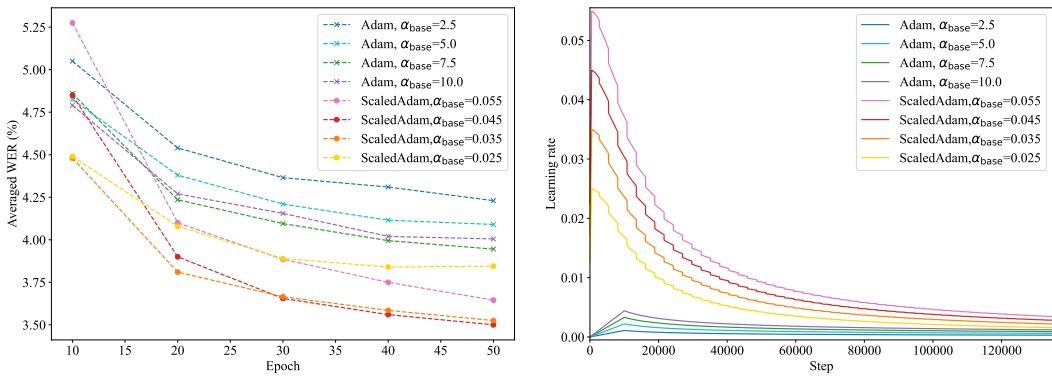

Figure A.2: Comparison between *ScaledAdam* and Adam in terms of: (Left) averaged WER on LibriSpeech *test-clean* and *test-other* at different epochs; (Right) learning rate at different steps.

## A.2 Activation functions

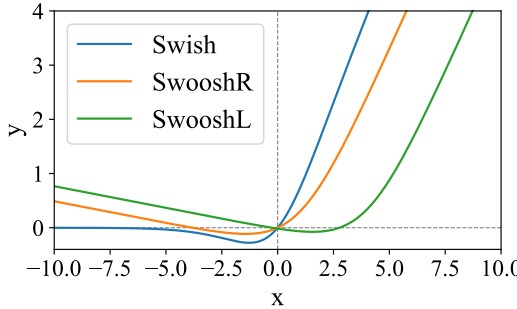

Figure A.3: The activation functions: Swish, *SwooshR*, and *SwooshL*.

## A.3 Activation constraints

Table 6: Ablation studies of activation constraints for *Zipformer*-M on LibriSpeech dataset. All models are trained for 40 epochs.

| Ablation | test-clean (%) | test-other (%) |
|---|---|---|
| *Zipformer*-M | 2.21 | 4.91 |
| No *Balancer* | 2.23 | 4.97 |
| No *Whitener* | 2.25 | 5.15 |
| No *Balancer*, No *Whitener* | 2.25 | 5.07 |

To ensure consistency in training and avoid badly trained modules, we propose the *Balancer* and the *Whitener*, which regularize the activations in a memory-efficient way. In forward pass, they are no-ops; in backward pass, they compute the gradients of the additional losses putting constraints on the activations $\mathbf{g}'$, and add that to the origin activation gradients $\mathbf{g}$: $\mathbf{g} = \mathbf{g} + \mathbf{g}'$. The placements of *Balancer* and *Whitener* may not seem to follow any very clear rules. They are generally applied when encountering specific instances of not-normally-trained models or model divergence. We first locate the abnormality to a specific module and then add the *Balancer* or *Whitener* to fix it.

### A.3.1 Balancer

Two failure modes commonly observed from the channel-wise statistical distribution are: **1)** the issue of activation values becoming excessively large or small can give rise to instability during the training process, particularly when employing mixed precision training; **2)** a significant number of "dead" neurons, whose outputs consistently remain negative, was observed upon examining the channel-wise statistics prior to the application of the non-linear activation function within the feed-forward modules. *Balancer* solves these issues by enforcing four constraints on each channel: lower and upper bounds of the mean value of absolute values, denoted as $a_{\min}$ and $a_{\max}$; minimum and maximum proportion of positive values, denoted as $p_{\min}$ and $p_{\max}$ respectively. Given the activation $\mathbf{x}$, intuitively we have $\mathrm{E}[\mathbf{x}] \propto \lambda$, where $\lambda$ represents the proportion of positive values. Due to the non-differentiable nature of positive value counting operation, a shifted version of Gaussian error function $\mathrm{erf}$ is introduced in order to approximate the mapping between $\mathrm{E}[\mathbf{x}] \in (-\infty, \infty)$ and $\lambda \in [0, 1]$ by $2 \cdot \mathrm{erf}(\mathbf{x}) - 1$, the inverse function of which can be approximated using $f_{\mathrm{pos} \to \mathrm{E}/\sqrt{\mathrm{Var}}}(x) = \mathrm{arctanh}(2x - 1)/(\sqrt{\pi} \cdot \log 2)$ without loss of generality. $\mu_{\min} = f_{\mathrm{pos} \to \mathrm{E}/\sqrt{\mathrm{Var}}}(p_{\min})$ and $\mu_{\max} = f_{\mathrm{pos} \to \mathrm{E}/\sqrt{\mathrm{Var}}}(p_{\max})$ can be further derived from the approximation. Following the same Gaussian assumption, the RMS is given by $\int_{-\infty}^{\infty} \frac{\sigma^2}{\sqrt{2\pi}} e^{-\frac{1}{2}(\mathbf{x}-\mu)^2} \mathrm{abs}(\mathbf{x})d\mathbf{x}$, where $\mu$ and $\sigma^2$ refer to the mean and variance of the Gaussian distribution. It can be approximated using $f_{\mathrm{abs} \to \mathrm{RMS}}(x) = \sqrt{\pi/2} \cdot x$ when $\mu \to 0$. Thus $r_{\min} = f_{\mathrm{abs} \to \mathrm{RMS}}(a_{\min})$ and $r_{\max} = f_{\mathrm{abs} \to \mathrm{RMS}}(a_{\max})$ can be further derived.

Specifically, the additional loss $\mathcal{L}_{\text{balancer}}$ conditioned on these constraints is defined as:

$$\mathcal{L}_{\text{RMS}} = |\log(\min(\max(\text{RMS}[\mathbf{x}], r_{\max}), r_{\min})/\text{RMS}[\mathbf{x}])|,$$

$$\mathcal{L}_{\text{E}/\sqrt{\text{Var}}} = |\text{E}[\mathbf{x}]/\sqrt{\text{Var}[\mathbf{x}]} - \text{clamp}(\text{E}[\mathbf{x}]/\sqrt{\text{Var}[\mathbf{x}]}, \mu_{\min}, \mu_{\max})|, \quad (9)$$

$$\mathcal{L}_{\text{balancer}} = \mathcal{L}_{\text{RMS}} + \mathcal{L}_{\text{E}/\sqrt{\text{Var}}},$$

where the statistics $\text{RMS}[\mathbf{x}]$, $\text{E}[\mathbf{x}]$, and $\sqrt{\text{Var}[\mathbf{x}]}$ are calculated in each channel. Before adding the additional gradient $\mathbf{g}' = \nabla_{\mathbf{x}}\mathcal{L}_{\text{balancer}}$ to the original activation gradient $\mathbf{g}$, $\mathbf{g}'$ is scaled to $\mathbf{g}' = \mathbf{g}' \cdot \alpha/\text{RMS}[\mathbf{g}'] \cdot |\mathbf{g}|$. Herein, $\alpha$ is used to prevent $\mathbf{g}'$ from overwhelming $\mathbf{g}$, and the per-element magnitude $|\mathbf{g}|$ is used to prevent the model from concentrating its "fixes" to the data distribution in frames with small gradients such as the padding frames. We set $\alpha = 0.04$ in this work.

### A.3.2 WHITENER

Another failure mode on activations is that for the feature covariance, one or a few eigenvalues are dominating while others are extremely small. This tends to happen in a model that is about to diverge. *Whitener* encourages a more informative output distribution, by restricting the feature covariance after mean subtraction to have less unequal eigenvalues. Specifically, for output $\mathbf{x} \in \mathcal{R}^{N \times D}$ with $N$ frames of $D$-dimensional features, we first compute the covariance matrix $C = (\mathbf{x} - \text{E}[\mathbf{x}])^T(\mathbf{x} - \text{E}[\mathbf{x}])$, where $C \in \mathcal{R}^{D \times D}$, and $\text{E}[\mathbf{x}]$ is per-channel mean. The auxiliary loss which measures the whiten metric $\mathcal{L}_{\text{whiten}}$ is defined as:

$$\mathcal{L}_{\text{whitener}} = (\sum_i \lambda_i^2/D)/(\sum_i \lambda_i/D)^2 = (\sum_i \sum_j C_{i,j}^2/D)/(\sum_i C_{i,i}/D)^2, \quad (10)$$

where $\boldsymbol{\lambda} = \{\lambda_1, \ldots, \lambda_D\}$ are the eigenvalues of the covariance matrix $C$. To keep the original activation gradient $\mathbf{g}$ dominant after adding the additional gradient $\mathbf{g}' = \nabla_{\mathbf{x}}\mathcal{L}_{\text{whitener}}$, $\mathbf{g}'$ is scaled to $\mathbf{g}' = \mathbf{g}' \cdot \alpha/\ell^2(\mathbf{g}') \cdot \ell^2(\mathbf{g})$, where $\ell^2$ denotes the L2 norm, and $\alpha$ is set to 0.01. The modification $\mathbf{g} = \mathbf{g} + \mathbf{g}'$ is done only when the whiten metric $\mathcal{L}_{\text{whiten}}$ is above a certain value $w_{\min}$ to prevent the model from learning pathological activation distributions. We usually set $w_{\min}$ to 10.

### A.3.3 ABLATION STUDIES.

We perform ablation experiments on LibriSpeech dataset to validate the effect of *Balancer* and *Whitener*. Table 6 presents the experimental results. All models are trained for 40 epochs. Removing *Balancer* does not lead to obvious change on model performance. However, it would increase the risk of model divergence without the value range constraints especially when employing mixed precision training. Removing *Whitener* results in 0.04% and 0.24% WER reduction on *test-clean* and *test-other*, respectively. This indicates that restricting the feature covariance to have less unequal eigenvalues in *Whitener* can boost performance.

## A.4 EXPERIMENTS ON LIBRISPEECH DATASET

### A.4.1 TRAINING CONFIGURATIONS OF ZIPFORMER MODELS

Before training, the Mel filter-bank features are per-computed and saved to disk. In training, we use *DynamicBucketingSampler* in Lhotse toolkit (Żelasko et al., 2021) to form the batches, where the batch size is determined dynamically given the constraint of the maximum total speech duration (in seconds). Table 7 presents the training configurations of *Zipformer* models on LibriSpeech dataset with speed perturbation with factors of 0.9, 1.0, and 1.1.

### A.4.2 COMPARISON WITH STATE-OF-THE-ART MODELS

As an extension of Table 2, Table 8 adds the results on LibriSpeech dataset for *Zipformer* with CTC and CTC/AED architectures respectively. For the *Zipformer* CTC/AED model, we use a 6-layer Transformer as AED decoder, each layer with attention dimension of 512, attention heads number of 8, and feed-forward hidden dimension of 2048. The *Zipformer* CTC models are trained for 100 epochs while the *Zipformer* CTC/AED models are trained for 50 epochs. Detailed training configurations are provided in Section A.4.1.

Table 7: Training configurations of *Zipformer* models on LibriSpeech dataset.

| Model | Type | Params (M) | Max duration (s) | GPUs | Epochs | Training time / epoch (m) |
|---|---|---|---|---|---|---|
| *Zipformer*-S | CTC | 22.1 | 1700 | 2 32G Tesla V100 | 100 | 86 |
| *Zipformer*-M | CTC | 64.3 | 1400 | 4 32G Tesla V100 | 100 | 60 |
| *Zipformer*-L | CTC | 147.0 | 1200 | 4 32G Tesla V100 | 100 | 76 |
| *Zipformer*-S | CTC/AED | 46.3 | 1700 | 2 32G Tesla V100 | 50 | 105 |
| *Zipformer*-M | CTC/AED | 90.0 | 1200 | 4 32G Tesla V100 | 50 | 67 |
| *Zipformer*-L | CTC/AED | 174.3 | 1200 | 4 32G Tesla V100 | 50 | 84 |
| *Zipformer*-S | pruned transducer | 23.3 | 1500 | 2 32G Tesla V100 | 50 | 87 |
| *Zipformer*-M | pruned transducer | 65.6 | 1000 | 4 32G Tesla V100 | 50 | 69 |
| *Zipformer*-L | pruned transducer | 148.4 | 1000 | 4 32G Tesla V100 | 50 | 80 |
| *Zipformer*-L | pruned transducer | 148.4 | 2200 | 8 80G Tesla A100 | 200 | 18 |

For the CTC systems, *Zipformer*-M outperforms Squeezeformer-ML on both test sets with only about half the number of parameters, and *Zipformer*-L also surpasses Squeezeformer-L by 0.27% on *test-other* with fewer parameters. For CTC/AED systems, *Zipformer*-M outperforms Conformer models and Branchformer, while *Zipformer*-L achieves comparable results with E-Branchformer-L. Note that as presented in Figure 3, *Zipformer*-L is much more efficient than E-Branchformer-L.

Table 8: WER(%) comparison between different models on LibriSpeech dataset. We also include the number of parameters and FLOPs of encoder for a 30s input audio measured with Deep-Speed (Rasley et al., 2020). *Trained with 8 80G NVIDIA Tesla A100 GPUs for 170 epochs.

| Model | Type | Params (M) | GFLOPs | *test-clean* (%) | *test-other* (%) |
|---|---|---|---|---|---|
| Squeezeformer-XS (Kim et al., 2022) | CTC | 9.0 | 18.2 | 3.74 | 9.09 |
| Squeezeformer-S (Kim et al., 2022) | CTC | 18.6 | 33.7 | 3.08 | 7.47 |
| Squeezeformer-SM (Kim et al., 2022) | CTC | 28.2 | 47.6 | 2.79 | 6.89 |
| Squeezeformer-M (Kim et al., 2022) | CTC | 55.6 | 88.4 | 2.56 | 6.50 |
| Squeezeformer-ML (Kim et al., 2022) | CTC | 125.1 | 183.3 | 2.61 | 6.05 |
| Squeezeformer-L (Kim et al., 2022) | CTC | 236.3 | 333.7 | 2.47 | 5.97 |
| E-Branchformer-B (Kim et al., 2023) | CTC/AED | 41.1 | 78.1 | 2.49 | 5.61 |
| Branchformer (Peng et al., 2022) | CTC/AED | 116.2 | 238.3 | 2.4 | 5.5 |
| E-Branchformer-L (Kim et al., 2023) | CTC/AED | 148.9 | 284.4 | 2.14 | 4.55 |
| Conformer-S (Gulati et al., 2020) | transducer | 10.3 | − | 2.7 | 6.3 |
| Conformer-M (Gulati et al., 2020) | transducer | 30.7 | − | 2.3 | 5.0 |
| Conformer-L (Gulati et al., 2020) | transducer | 118.8 | − | 2.1 | **4.3** |
| Conformer in WeNet (Zhang et al., 2022b) | CTC/AED | 121.3 | − | 2.66 | 6.53 |
| Conformer in ESPnet (Miyazaki et al., 2023) | CTC/AED | 113.2 | − | 2.29 | 5.13 |
| Conformer-S | pruned transducer | 9.8 | 29.1 | 3.75 | 9.24 |
| Conformer-M | pruned transducer | 28.4 | 77.0 | 2.96 | 7.11 |
| Conformer-L | pruned transducer | 122.5 | 294.2 | 2.46 | 5.55 |
| *Zipformer*-S | CTC | 22.1 | 40.8 | 2.85 | 6.91 |
| *Zipformer*-M | CTC | 64.3 | 62.9 | 2.51 | 6.02 |
| *Zipformer*-L | CTC | 147.0 | 107.7 | 2.49 | 5.7 |
| *Zipformer*-S | CTC/AED | 46.3 | 40.8 | 2.46 | 6.04 |
| *Zipformer*-M | CTC/AED | 90.0 | 62.9 | 2.22 | 4.97 |
| *Zipformer*-L | CTC/AED | 174.3 | 107.7 | 2.09 | 4.59 |
| *Zipformer*-S | pruned transducer | 23.3 | 40.8 | 2.42 | 5.73 |
| *Zipformer*-M | pruned transducer | 65.6 | 62.9 | 2.21 | 4.79 |
| *Zipformer*-L | pruned transducer | 148.4 | 107.7 | 2.06 | 4.63 |
| *Zipformer*-L* | pruned transducer | 148.4 | 107.7 | **2.00** | 4.38 |

