# OpenReview forum: "Zipformer: A faster and better encoder for automatic speech recognition"
_ICLR.cc/2024/Conference — ICLR 2024 oral_

### Official Review · Reviewer_hvPD · 2023-10-26

**Soundness:** 3 good
**Presentation:** 3 good
**Contribution:** 4 excellent
**Rating:** 8
**Confidence:** 4

**Summary:**

This paper offers a number of incremental improvements over the well-known Conformer architecture. In aggregate, these improvements set a new state of the art for supervised Librispeech.

**Strengths:**

This paper makes the most significant type of contribution in ASR - it improves the state of the art on a well-studied benchmark.  Since it has done so with a collection of small improvements, it offers detailed ablations quantifying the usefulness of each modification.  Importantly, it appears that each modification by itself offers improvement over Conformer.

**Weaknesses:**

In my option, this paper has the same problem that it identifies in the original Conformer write-up: It's closed-source, complicated, and it will likely not be possible for the results to be reproduced.  This is not by itself disqualifying; it is the nature of the field that SOTA systems are often published without source code.  However, I expect that future work iterating on this design will struggle with direct comparison, much in the same way that this paper compares to a weaker reproduction of the original Conformer.

**Questions:**

The comparison to the original conformer is not quite apples-to-apples, in that Zipformer-L is substantially larger than Conformer-L in both the original publication and in this paper's reproduction.  I think the authors should consider including a variant of Zipformer with ~120M parameters, as originally used by Gulati et al.   This would prevent the suspicion that the stronger WER simply reflects the fact that the Zipformer is larger.

Still, I think Zipformer's substantially reduced computational needs demonstrate that superiority over Conformer does not only reflect model size.  So, I'll recommend this paper for acceptance despite the flaw in this comparison.

---

> ### Comment · Reviewer_6jVU · 2023-11-13
> **State-of-the-art Librispeech**
>
> I just want to leave a short comment on this:
>
> > these improvements set a new state of the art for supervised Librispeech
>
> This is only partly true.
>
> First of all, the standard Librispeech task is usually together with a language model. Together with a language model, as far as I know, the best result is 3.65% WER on test-other with the E-Branchformer (https://arxiv.org/abs/2210.00077).
>
> Ok, but it's also legitimate to specifically look at results without external language model, as the authors do here. But in that case, they also don't quite reach state-of-the-art on test-other, which is still by the original Conformer-L with 4.3%, better than 4.38% which they reach here.
>
> On test-clean, without language model, yes, I think they reached state-of-the-art. But this is maybe the not-so-interesting-case.
>
> Nonetheless, the results they present are still very good, and specifically their relative improvements.
>
> But more critically speaking, you could also say, their Conformer baseline with  5.55% WER on test-other is a bit weak baseline compared to other standard Conformer results, e.g. the 4.3% by Gulati et al., 2020, or 4.74% in https://arxiv.org/abs/2210.00077.

---

> > ### Comment · Reviewer_hvPD · 2023-11-15
> > **correct**
> >
> > This point is well taken.  I think we still agree that the presented results are quite strong.

---

> > ### Author Response · Authors · 2023-11-21
> > **Thanks and response to concerns**
> >
> > Thanks for your detailed review.
> >
> > We tried LM rescoring on the Zipformer-L model with pruned-transducer. The LM contains 18 transformer layers, with dimension of 768 and 100M parameters, trained on the LibriSpeech langauge model corpus and the LibriSpeech 960h transcripts. We select 100 best paths from the transducer beam search results for rescoring with the LM. It gets comparable results with the Conformer paper, which is also a transducer system.
> > - w/o LM rescoring: Conformer-L, 2.1/4.3; Zipformer-L, 2.00/4.38
> > - w/ LM rescoring: Conformer-L, 1.9/3.9; Zipformer-L, 1.85/3.92
> >
> > It is worse than the results of 1.81/3.65 from E-Branchformer-L with Internal Language Model Estimation (ILME)-based rescoring (https://arxiv.org/pdf/2210.00077.pdf). However, since this paper focuses on improving the encoder model, we report the results without external LM to fairly compare the modeling capabilities of different encoders. We leave the lm rescoring with advanced methos (e.g., shallow fusion, Low-Order Density Ratio, ILME) for future research.

---

> ### Author Response · Authors · 2023-11-20
> **Thanks and response to concerns**
>
> Thank you for your detailed review, below we address the key concerns.
>
> > In my option, this paper has the same problem that it identifies in the original Conformer write-up: It's closed-source, complicated, and it will likely not be possible for the results to be reproduced. This is not by itself disqualifying; it is the nature of the field that SOTA systems are often published without source code. However, I expect that future work iterating on this design will struggle with direct comparison, much in the same way that this paper compares to a weaker reproduction of the original Conformer.
>
> Our code is open-sourced. The link will be put in the final version.
>
> > The comparison to the original conformer is not quite apples-to-apples, in that Zipformer-L is substantially larger than Conformer-L in both the original publication and in this paper's reproduction. I think the authors should consider including a variant of Zipformer with ~120M parameters, as originally used by Gulati et al. This would prevent the suspicion that the stronger WER simply reflects the fact that the Zipformer is larger. Still, I think Zipformer's substantially reduced computational needs demonstrate that superiority over Conformer does not only reflect model size. So, I'll recommend this paper for acceptance despite the flaw in this comparison.
>
> We thank the reviewer for the observation. We would like to suggest that the majority of the experiments are all conducted using M-scaled Zipformer model, hence the difference of # params on L-scaled model between Zipformer and Conformer does not invalidate the conclusions we claimed in the manuscript. When it comes to Zipformer-L, we would like to slightly extend the boundary of where it could reach to given its efficiency in resource usage and computational time.

---

> > ### Comment · Reviewer_hvPD · 2023-11-21
> > **Acknowledged**
> >
> > Thank you for the response! I'm glad to hear that the code will be released with the final version.  I'll leave my rating at 8.

---

### Official Review · Reviewer_2d3G · 2023-11-01

**Soundness:** 3 good
**Presentation:** 3 good
**Contribution:** 3 good
**Rating:** 6
**Confidence:** 5

**Summary:**

This paper presents an interesting alternative, zipformer, to the transformer/conformer encoder which are widely used for automatic speech recognition. The author also claims their modified optimizer, ScaledAdam, learns faster and converges to better optimum than the standard adam. The authors have presented that with the proposed zipformer and together with the modified optimizer, the resultant system achieves similar performance compared to conformer, but with better memory and FLOPS efficiency.

**Strengths:**

This work presents an alternative approach to the standard (or widely used) transformer encoder structure, which is very interesting to read. The authors explain motivations behind some of the proposed modification, together with the ablation studies.

**Weaknesses:**

From my biased view, the major weakness in this paper lies in the fact that this work actually presents two inter-connected but (arguably) separate works: one is to the novel new encoder structure, including the U-net with middle stacks operate at a lower frame rates, sharing the attention weight with two self attentions, a novel non-linear attention, a BiasNorm and a slightly modified swooshL/swooshR activation function; the other is about the modified Adam optimizer, scaledAdam, which the authors claim that it can explicitly learn the parameter scale, which the widely used adam failed to. Though the author gives some motivations behind these changes and give some ablation studies, it still points to the following unanswered questions:

  - the authors claim that the zipformer is better than the (reproduced) conformer, squeezeformer or other variants of transformer. However, it is unclear to me whether this is because of the model structure or because of the modified optimizer ? For example, in Table 5, the author presents that with all the proposed change (model and optimizer), zipformer-M can achieve 4.79% WER on test-other, however, with the standard adam optimizer, the WER becomes 5.51%. Will the other variants of transformer get better WER with the author proposed optimizer ?

  - On the other hand, I am wondering whether the proposed optimizer, which has many adhoc modifications compared with the much widely used adam optimizer, can be more widely used for other deep learning optimization tasks or does it only work for speech tasks or does it only work for zipformer ? Given the zipformer-L is a pretty small model (only 60M parameters) trained on limited data (<1000hrs) according to today's standard,  will the proposed change still work better than the standard adam when the model and the training data scales up?


Other weakness of this paper include:

  - The authors propose changes to many widely used components, which have been well tested over time and over different tasks beyond just speech recognition. For example, the authors claim that the mean extraction in layer norm is not necessary and also remove the variance normalization. While this may seem to work for authors use case, I am intrigued to learn how generalizable this claim can be.

  - Some of the proposed modification seem arbitrary. It would be great if the authors can explain a bit. For example, in Eq (4), for the swooshR and swooshL functions, how the -0.08 is selected ? The authors mentioned that the coefficient 0.035 in Eq (4) is slightly tuned. How general is this parameter ?

- In the ablation study, quite a few modifications results in less than 0.2% absolute WER change. How sensitive of the WERs on librispeech relative to other factors like checkpoint choosing, different training run with different seed ? In my own experiments, these factors can result in up to 0.1% WER changes.


- The authors also miss a series of work in the literature that directly apply standard transformer (without much modification) for ASR. For example,
   * Y. Wang, A. Mohamed, D. Le, C. Liu et al., “Transformer-based Acoustic Modeling for Hybrid Speech Recognition,” in Proc ICASSP, 2020, it reports a WER of 2.26% / 4.85% on test-clean/test-other of librispeech, achieved by CTC + external NNLM;
This is followed by a work in:
  * Zhang, Frank, et al. "Faster, simpler and more accurate hybrid asr systems using wordpieces." arXiv preprint arXiv:2005.09150 (2020).
which achieves a WER of 2.10% and 4.20% using a similar architecture.

It is also noted that in

- Wang, Y., Chen, Z., Zheng, C., Zhang, Y., Han, W. and Haghani, P., 2023, June. Accelerating rnn-t training and inference using ctc guidance. In ICASSP 2023-2023 IEEE International Conference on Acoustics, Speech and Signal Processing (ICASSP) (pp. 1-5). IEEE.

It achieved almost the same WER as the initial conformer paper, but part of the conformer layers (10 out of total 17 layers) is running at much lower frame rate (about 120ms-160ms per frame). This is related to the temporal downsampling used in zipformer.

**Questions:**

- In Table 5 and section 4.0.3, it is unclear to me why "no temporal downsampling" will result in increased #params (from 65.6M to 94.2M). All the major components in zipformers: attention , feedfoward and convolution use the same parameters for different temporal resolution.

---

> ### Author Response · Authors · 2023-11-20
> **Thanks and response to concerns**
>
> Thank you for your detailed review, below we address the key concerns.
>
> > However, it is unclear to me whether this is because of the model structure or because of the modified optimizer? .
>
> As in Table 5, the new model structure, BiasNorm, Swoosh functions, and ScaledAdam all contribute to the performance. The model structure is mainly for higher efficiency. Swoosh functions and ScaledAdam have greater impact on performance improvement.
>
> > Will the other variants of transformer get better WER with the author proposed optimizer ?
>
> We validated ScaledAdam on LibriSpeech using a 12-layer Conformer  (attention dim=512, feed-forward dim=2048). Results indicate that ScaledAdam leads to faster convergence and better performance on Conformer.
>
> * Adam (epoch index, WER(%) on test-clean/test-other):
>
> epoch-10, 3.13/7.62; epoch-20, 2.83/6.88; epoch-30, 2.76/6.67;
>
> epoch-40, 2.79/6.62; epoch-50, 2.77/6.48; epoch-60, 2.74/6.43.
>
> * ScaledAdam:
>
> epoch-10, 3.15/7.35; epoch-20, 2.68/6.32; epoch-30, 2.55/6.12;
>
> epoch-40, 2.53/5.96; epoch-50, 2.55/6.02; epoch-60, 2.55/6.0.
>
> > I am wondering whether the proposed optimizer ... does it only work for speech tasks or does it only work for zipformer?
>
> We validated ScaledAdam and Swoosh on LM task. Results of LMs with 18 transformer layers using GeLU (#param. 98M) trained on the Libriheavy transcripts (https://arxiv.org/pdf/2309.08105.pdf) suggest they can improve the performance:
>
> Exp1: Adam w. CosineAnnealingLR and lr=4e-4;
>
> Exp2: ScaledAdam w. Eden and $\alpha_{\text{base}}=0.025$;
>
> Exp3: ScaledAdam w. Eden and $\alpha_{\text{base}}=0.025$, replace GeLU with SwooshL.
>
> Trained for 12000 iter with 1B tokens, the validation loss values (negative log-likelihood):
>
> Exp1: 0.947; Exp2: 0.9369; Exp3: 0.9229
>
> Trained for 24000 iter with 2B tokens, the validation loss values (negative log-likelihood):
>
> Exp1: 0.9068; Exp2: 0.885; Exp3: 0.8725
>
> > Given the zipformer-L is a pretty small model ... when the model and the training data scales up?
>
> As in Table 4, experiments on the 10000-hr WenetSpeech manifest that Zipformer-M/L outperform all other models on Test Net and Test Meeting sets.
>
> A 288M Zipformer on the 10000-hr GigaSpeech gets WERs of 10.07/10.19 on dev/test sets, while the 66M Zipformer-M gets 10.25/10.38. To the best of our knowledge, this is so far the lowest WER reported on GigaSpeech as on https://github.com/SpeechColab/GigaSpeech.
>
> > The authors propose changes to many widely used components, ..., I am intrigued to learn how generalizable this claim can be.
>
> We would like to emphasize the components are focused on ASR. Our experiments on LM task suggest that these can be generalized to other tasks to a certain extent, e.g., ScaledAdam and Swoosh. Please refer to the response of question 3.
>
> Empirically, the mean subtraction is used by systems with LayerNorm as a way to "defeat LayerNorm", by making certain elements very large and nearly constant. This is not necessary with BiasNorm because we can "defeat normalization" by having a large learnable bias. We normalize the vector by RMS instead of Var. We found an extra sqrt in the denominator in Eq.2 and removed that in the revised version.
>
> > Some of the proposed modification seem arbitrary. It would be great if the authors can explain a bit. For example, in Eq (4), for the swooshR and swooshL functions, how the -0.08 is selected ? The authors mentioned that the coefficient 0.035 in Eq (4) is slightly tuned. How general is this parameter?
>
> The -0.08 is chosen to make the negative slope to be 10% of the positive slope.
> The coefficient 0.035 was tuned on LibriSpeech and maintained the same on all other experiments including the LM ones, suggesting its capability of generalization (refer to the response of question 3).
>
> > How sensitive of the WERs on librispeech relative to other factors like checkpoint choosing, different training run with different seed?
>
> In our experiments, we search for different checkpoints for decoding and report the best result. In training, we set a fixed random seed for the libraries, such as random, numpy, and torch.
>
> > Missing references
>
> We have added these references in Section 2.
>
> > About the referred paper Accelerating rnn-t training and inference using ctc guidance
>
> We cited it in Section 2 as a related work. It leverages the guidance from the auxiliary CTC head to skip frames on the encoder output and even in the middle of the encoder. However, our work focus on improving the sequence encoder structure itself.
>
> > In Table 5 and section 4.0.3, it is unclear to me ... for different temporal resolution.
>
> As described in Section 3.1 and Section 4.0.1 (in Table 1), stacks are of various embedding dims and feed-forward dims. For the ablation study ``no temporal downsampling", we use Conv-Embed with downsampling rate of 4, and 12 Zipformer blocks with the same dimensions as in the middle stack (embedding dim=512, feed-forward dim=1536).

---

### Official Review · Reviewer_6jVU · 2023-11-01

**Soundness:** 3 good
**Presentation:** 2 fair
**Contribution:** 3 good
**Rating:** 8
**Confidence:** 5

**Summary:**

Context: Automatic speech recognition (ASR), where Conformer currently is the state-of-the-art encoder, which is used by most groups. A new variant of the Conformer is proposed in this paper, along with a lot of changes.

- New ZipFormer model as an alternative to the Conformer.
- BiasNorm as an alternative to LayerNorm.
- Bypass module instead of residual connections
- Downsample module instead of simply average or max pooling or striding
- Re-use of attention weights inside a ZipFormer block
- Non-linear attention instead of standard attention, which uses a multiplicative element
- New activation functions SwooshR and SwooshL as an alternative to Swish.
- Additional learnable scalar for the scale of weights
- ScaledAdam optimizer as an alternative to Adam.
- Eden learning rate scheduling which is both step-based and epoch-based, as an alternative to purely step-based or purely epoch-based LR schedules.

**Strengths:**

- There are a lot of interesting novelties here in the paper, like the ZipFormer model itself, ScaledAdam, BiasNorm, new activation functions, and more. (Although having so many different novelties is also a weakness, see below.)

- Improvements are very good, i.e. good relative WER improvements, while also having it more efficient.

- Good ASR baselines.

**Weaknesses:**

- There are maybe too many new things being introduced here, which are all interesting by themselves, but each of them would maybe require more investigation and analysis on their own. E.g. introducing a new optimizer (ScaledAdam) is interesting, but it should be tested on a couple of different models and benchmarks, and this would basically a work on its own. Now we have way too little analysis for each of the introduced methods to really tell how good they are. The ablation study is basically only a single experiment, showing ScaledAdam vs Adam, or LayerNorm vs BiasNorm. A single experiment is not really enough to really see how ScaledAdam vs Adam performs, or BiasNorm vs LayerNorm, etc. E.g. replaying LayerNorm by BiasNorm in a standard Transformer, maybe for language modeling, how would this perform? Many of these introduced methods seem to be quite orthogonal to each other.

- For comparisons, it would have been better to use a more standard transducer, not a pruned transducer. Or maybe even CTC, or CTC/AED. This would have made it easier to compare the results to other results from the literature (Table 2).

- Measuring just FLOPs can sometimes be misleading as indicator for speed, because certain operations might be faster than others, and certain operations can be executed in parallelized, while others can not. It would be interesting to also see the actual measured speed on some given hardware.

- No systematic comparison to other Conformer variants, e.g. E-Branchformer, etc. (I don't just mean to have the number from literature as a comparison, as done in Table 2. I mean that the differences are compared, i.e. discussed and maybe also experimentally compared.)

**Questions:**

The capitalization is a bit inconsistent. Transformer is always lower case (transformer), while Conformer, ZipFormer etc are all capitalized.

It would make sense to state that (or whether) the code is published (even if you want to put the link only in the final version).

How long does the training take? What batch sizes are used?


> Downsample module averages every 2 frames with 2 learnable scalar weights (after softmax normalization)

I don't exactly understand how this is done. For the whole module, there are two learnable scalar weights only? So it could learn to always take really the average, but also to always take the first and ignore the second, or vice versa? This does not make sense to me. Why does it make sense to learn that? What happens if you just take the average? Or maybe use max-pooling instead? What is the influence of this?


If you downsample at the very end to 25 Hz, why not do that earlier, and downsample right in the beginning to 25 Hz, and then never to 50 Hz again? That would make it quite a bit faster. So how much worse does this get?


LayerNorm/BiasNorm: When there is a large value, as even explicitly modelled by the bias in BiasNorm, isn't this a problem, that there is a large number in the denominator, so then all the values become very small? The exp(γ) can compensate that then again, however, I wonder if there are numerical problems.

What is the influence of exp(γ) vs just γ (without exp) in BiasNorm?


How does ScaledAdam compare to methods like WeightNorm, where the weights are also normalized?


> Since Adam is nearly invariant to changes in the gradient scale, for simplicity we replace this with ht = gt ·(rt−1 ⊙θt′−1) = gt ·θt−1

I'm not sure that this would make the code simpler, to have this special case? But I also wonder, what is the influence of this?


For ScaledAdam, as I understand, adding the learnable parameter scale would actually change the model, as this scale now is also part of the model? Is this added for every parameter, or only some selected ones, e.g. the linear transformations?

What is the influence of the Eden learning rate scheduling?

---

> ### Author Response · Authors · 2023-11-20
> **Thanks and response to concerns**
>
> Thank you for your detailed review, below we address the key concerns. Spelling issues are fixed in the revised version, link to the code will be attached in the final version.
>
> >  About the proposed BiasNorm, Swoosh, ScaledAdam on language modeling task
>
> We validated these components on LM task. Results indicate that using ScaleAdam and Swoosh function achieves better performance. A 18-layer transformer LM was built w. GeLU (# param. is 98M). The model was trained on the Libriheavy transcripts (https://arxiv.org/pdf/2309.08105.pdf).
>
> Exp1: Adam w. CosineAnnealingLR and lr=4e-4;
>
> Exp2: ScaledAdam w. Eden and $\alpha_{\text{base}}=0.025$;
>
> Exp3: ScaledAdam w. Eden and $\alpha_{\text{base}}=0.025$, replace GeLU with SwooshL.
>
> Trained for 12000 iter with 1B tokens, the validation loss values (negative log-likelihood):
>
> Exp1: 0.947; Exp2: 0.9369; Exp3: 0.9229
>
> Trained for 24000 iter with 2B tokens, the validation loss values (negative log-likelihood):
>
> Exp1: 0.9068; Exp2: 0.885; Exp3: 0.8725
>
> However, BiasNorm causes slightly worse result. It is possible that the BiasNorm might not work for LM task, since our modification are done based on the observation on ASR experiments. The ablation study in Tab. 5 manifests that it is effective on Zipformer models.
>
> > For comparisons, it would have been better to use a more standard transducer, ... would have made it easier to compare the results to other results from the literature (Table 2).
>
> We updated results of Zipformer models with CTC and CTC/AED systems in Appx. Sec. A.4.2 in the revised version. Overall, Zipformer models still outperform other ASR encoders.
>
> > The actual measured speed on some given hardware.
>
> We compared the speed and memory usage between different ASR encoders in inference mode on a 32G V100 in Figure 3. In overall, Zipformer achieves a better trade-off between performance and efficiency than other encoders.
>
> > No systematic comparison to other Conformer variants
>
> Besides the performance comparison in Table 2, Figure 3 compares their efficiency in terms of speed and memory usage, and Sec. 2 describes their architectures and main difference between Zipformer and these variants.
>
> > How long does the training take? What batch sizes are used?
>
> Please refer to Appx. Sec. A.4.1 of the revised version.
>
> > "Downsample module ... What happens if you just take the average? Or maybe use max-pooling instead?
>
> With a factor of 2, the Downsample just performs the weighted average on every 2 frames with 2 learnable scalars that sum to one. This slightly outperforms average-pooling and max-pooling, on LibriSpeech test-clean/test-other:
>
> * ours, 2.21/4.79
>
> * average, 2.18/4.96
>
> * max-pooling, 2.24/4.87
>
> > If you downsample at the very end to 25 Hz, why not do that earlier ... So how much worse does this get?
>
> This degrades the performance of Zipformer-M from 2.21/4.79 to 2.45/5.43 on LibriSpeech test-clean/test-other. This level of downsampling at the middle stack might be too aggressive to cause information loss.
>
> > LayerNorm/BiasNorm: When there is a large value, ..., I wonder if there are numerical problems.
>
> It wouldn't be so large as to cause numerical problems. It just needs large enough to dominate the denominator so that the some length information could be retained, e.g., 50 in LayerNorm.
>
> > What is the influence of exp(γ) vs just γ (without exp) in BiasNorm?
>
> Learning the scale in log space is safer in theory, since it avoids the gradient oscillation problem that we describe in Sec. 3.3.
>
> > How does ScaledAdam compare to methods like WeightNorm, where the weights are also normalized? For ScaledAdam, as I understand, adding the learnable parameter scale would actually change the model, as this scale now is also part of the model? Is this added for every parameter, or only some selected ones, e.g. the linear transformations?
>
> WeightNorm decouples the magnitude of a parameter from its direction, resulting in two parameters. ScaledAdam does not learn an extra parameter for the scale. It learns the scale by adding an update term (in Eq. 7), which amounts to adding an extra gradient term in the direction of rescaling each parameter. It eliminates the need to specify the modules to apply the weight normalization.
>
> We mentioned this in Sec. 3.5 of the revised version.
>
> > I'm not sure that this would make the code simpler, to have this special case? But I also wonder, what is the influence of this?
>
> This makes the code simpler as we don't need to compute $\boldsymbol\theta_{t-1}'$, which makes almost no impact since Adam's update magnitudes are invariant to gradient rescaling.
>
> > What is the influence of the Eden learning rate scheduling?
>
> Eden is specially designed for ScaledAdam. Since ScaledAdam is more stable than Adam, we don't need a long warm-up period like in Conformer's learning rate schedule. We also use larger learning rate values, because we scale the update by parameter RMS (in Eq. 6), which is normally much smaller than one.

---

### Official Review · Reviewer_UyUP · 2023-11-04

**Soundness:** 3 good
**Presentation:** 3 good
**Contribution:** 3 good
**Rating:** 8
**Confidence:** 4

**Summary:**

This paper proposes a new encoder architecture based on conformer for speech recognition task. Some of the newly introduced modifications are well motivated based on some findings on trained the original conformer models. The experiments show strong performance compared with original conformer model and multiple other conformer variants. Considering the popularity of conformer for speech recognition and other speech tasks, the findings and contribution of this paper could benefit the speech/audio community a lot!

**Strengths:**

1. A newly designed conformer variant that achieve SOTA performance on speech recognition task.
2. Experiments are done on different datasets with training data at different scales (hundreds, thousands and tens of thousands).
3. Experimental results are strong, indicating the effectiveness of model.

**Weaknesses:**

1. While the motivation for biasnorm and scaledadam are well explained, the motivation for zipformer blcok, especially those downsampling and upsampling modules are not well presented.
2. The results on aishell1 is not quite convincing compared to other conformer variants. The author could elaborate more on the performance. Could be that this is a small dataset?

**Questions:**

1. Curious about whether the authors tried the new activation functions, biasnorm and scaledadam on other tasks? Do they still show superiority?
2. How could you make the model work in streaming fasion?

---

> ### Author Response · Authors · 2023-11-20
> **Thanks and response to concerns**
>
> Thank you for your detailed review, below we address the key concerns.
>
> > While the motivation for biasnorm and scaledadam are well explained, the motivation for zipformer blcok, especially those downsampling and upsampling modules are not well presented.
>
> Zipformer uses a downsampled encoder structure which processes the sequence at various frame rates. It allows to efficiently learn representation at different temporal resolutions. According to the comment, we have made this motivation more clear in the revised version.
>
> As described in Section 3.2, Zipformer block has about twice the depth of the Conformer block, reusing the attention weights in different module to save computation and memory.
>
> > The results on aishell1 is not quite convincing compared to other conformer variants. The author could elaborate more on the performance. Could be that this is a small dataset?
>
> Yes, Aishell-1 is a small dataset containing only 170 hours of speech, which is described in Section 4.0.1. Note that as presented in Table 3, Zipformer-S achieves better performance than Conformer implemented in ESPnet toolkit with fewer parameters, and scaling up Zipformer achieves better performance.
>
> > Curious about whether the authors tried the new activation functions, biasnorm and scaledadam on other tasks? Do they still show superiority?
>
> We tried Swoosh function, BiasNorm, and ScaledAdam on language modeling task. Experimental results indicate that ScaledAdam and Swoosh function are able to improve the performance, while BiasNorm can't. Specifically, we build an 98M language model with 18 transformer layers with GeLU activation. The model is trained on the transcripts of Libriheavy dataset (https://arxiv.org/pdf/2309.08105.pdf). The transcripts are converted into 500-class BPE tokens. We conduct following experiments:
>
> **Exp1**: use Adam optimizer with CosineAnnealingLR schedule and lr=4e-4;
>
> **Exp2**: use ScaledAdam optimizer with Eden schedule and $\alpha_{\text{base}}=0.025$;
>
> **Exp3**: use ScaledAdam optimizer with Eden schedule and $\alpha_{\text{base}}=0.025$, replace GeLU with SwooshL.
>
> When trained for 12000 iterations with 1B tokens, the validation loss values (negative log-likelihood) are:
>
> **Exp1**: 0.947; **Exp2**: 0.9369; **Exp3**: 0.9229
>
> When trained for 24000 iterations with 2B tokens,  the validation loss values (negative log-likelihood) are:
>
> **Exp1**: 0.9068; **Exp2**: 0.885; **Exp3**: 0.8725
>
> > How could you make the model work in streaming fasion?
>
> The way to make conformer streaming also applies to Zipformer. Specifically, for attention modules, in training we could limit the future contexts by applying the block-triangular masks on attention weights. We should also replace regular convolutions with causal convolutions. In inference, we could cache the left contexts for attention modules and convolution modules.

---

### Author Response · Authors · 2023-11-20
**Thanks for the reviews and summary of key paper changes**

We thank the reviewers for their detailed and thoughtful reviews. The reviewers all appreciated the contribution of this paper with rating scores of 8,8,6,8.

Detailed responses are provided in individual responses to each review.  We summarize the key changes made in the revised version bellow:

* Add the results of Zipformer models with CTC and CTC/AED systems on LibriSpeech dataset in Appendix Section A.4.2. For these systems, Zipformer model still outperforms other encoders.
* More detailed training configutations of Zipformer models on LibriSpeech dataset is presented in Appendix Section A.4.1.
* Add discussion of the difference between ScaledAdam and weight normalization in Section 3.5.
* Figure 4 is moved to Appendix Figure A.2 for a more detailed demonstration.
* An extra sqrt in Equation 2 is found and removed.
* Add missing references.

---

### Meta-Review · Area_Chair_yo6N · 2023-12-02

**Metareview:**

In this work, the authors present an a number of incremental improvements over the well-known Conformer architecture attaining a new state of the art for supervised Librispeech. A good experimental validation is presented.

**Justification For Why Not Higher Score:**

N/A

**Justification For Why Not Lower Score:**

Newly introduced modifications are well motivated, and experimental results are sound.

---

### Decision · Program_Chairs · 2024-01-16

Accept (oral)